# Increased Oral Care Needs and Third Molar Symptoms in Women with Gestational Diabetes Mellitus: A Finnish Gestational Diabetes Case–Control Study

**DOI:** 10.3390/ijerph191710711

**Published:** 2022-08-28

**Authors:** Jenni Pukkila, Sanna Mustaniemi, Shilpa Lingaiah, Olli-Pekka Lappalainen, Eero Kajantie, Anneli Pouta, Risto Kaaja, Johan G. Eriksson, Hannele Laivuori, Mika Gissler, Marja Vääräsmäki, Elina Keikkala

**Affiliations:** 1PEDEGO Research Unit, Medical Research Centre Oulu, Oulu University Hospital, University of Oulu, 90220 Oulu, Finland; 2Population Health, Public Health and Welfare, Finnish Institute for Health and Welfare, 00271 Helsinki and 90220 Oulu, Finland; 3Department of Oral and Maxillofacial Diseases, Helsinki University Hospital, University of Helsinki, 00290 Helsinki, Finland; 4Children’s Hospital, Helsinki University Hospital, University of Helsinki, 00290 Helsinki, Finland; 5Department of Clinical and Molecular Medicine, Norwegian University of Science and Technology, 7028 Trondheim, Norway; 6Department of Government Services, Finnish Institute for Health and Welfare, 00271 Helsinki, Finland; 7Internal Medicine, Institute of Clinical Medicine, Turku University Hospital, University of Turku, 20521 Turku, Finland; 8Technology and Research, Agency for Science, Singapore Institute for Clinical Sciences, Singapore 117609, Singapore; 9Department of Obstetrics and Gynaecology and Human Potential Translational Research Programme, Young Loo Lin School of Medicine, National University of Singapore, Singapore 119228, Singapore; 10Department of General Practice and Primary Health Care, Helsinki University Hospital, University of Helsinki, 00290 Helsinki, Finland; 11Folkhälsan Research Center, 00250 Helsinki, Finland; 12Adolescent and Maternal Health Research, Center for Child, Faculty of Medicine and Health Technology, Tampere University, 33520 Tampere, Finland; 13Department of Obstetrics and Gynaecology, Tampere University Hospital, 33520 Tampere, Finland; 14Medical and Clinical Genetics, Helsinki University Hospital, University of Helsinki, 00290 Helsinki, Finland; 15Institute for Molecular Medicine Finland, Helsinki Institute of Life Science, University of Helsinki, 00290 Helsinki, Finland; 16Department of Knowledge Brokers, Finnish Institute for Health and Welfare, 00271 Helsinki, Finland; 17Department of Molecular Medicine and Surgery, Karolinska Institute, 104 35 Stockholm, Sweden; 18Region Stockholm, Academic Primary Health Care Centre, 113 65 Stockholm, Sweden

**Keywords:** diabetes, gestational, oral health, women’s health, molar, third, self-report

## Abstract

(1) Hyperglycemia and oral pathology accelerate each other in diabetes. We evaluated whether gestational diabetes mellitus (GDM) is associated with self-reported increased oral health care needs and oral symptoms, including third molar symptoms, during pregnancy. (2) Pregnant women with (*n* = 1030) and without GDM (*n* = 935) were recruited in this multicenter Finnish Gestational Diabetes study in 2009–2012. Of the women with GDM, 196 (19.0%) receiving pharmacological treatment, 797 (77.0%) receiving diet treatment and 233 (23.0%) with recurrent GDM were analyzed separately. Oral health was assessed using structured questionnaires and analyzed by multivariable logistic regression adjusted for background risk factors. (3) Women with GDM were more likely to report a higher need for oral care than controls (31.1% vs. 24.5%; odds ratio (OR) 1.39; 95% confidence interval (CI) 1.14–1.69), particularly women with recurrent GDM (38.1% vs. 24.5%; OR 1.90; 95% CI 1.40–2.58). Women with pharmacologically treated GDM (46.9%) more often had third molar symptoms than controls (36.1%; OR 1.57; 95% CI 1.15–2.15) than women with diet-treated GDM (38.0%; OR 1.47; 95% CI 1.07–2.02). (4) GDM is associated with perceived oral care needs. Third molar symptoms were associated with pharmacologically treated GDM.

## 1. Introduction

Gestational diabetes mellitus (GDM) is defined as elevated blood glucose levels observed for the first time during pregnancy (World Health Organization criteria) [1]. The prevalence of GDM worldwide has increased over the last decade—with reported rates of up to 25%—owing in part to the implementation of comprehensive screening programs. In addition, the major risk factors for GDM—advanced age and obesity—have become more frequent among pregnant women [2,3,4]. GDM is considered a prediabetic stage, because approximately 50% later develop type 2 diabetes [3]. Women with recurrent GDM or requiring pharmacological treatment for GDM are at especially high risk of later developing type 2 diabetes [5,6].

Periodontal disease is a group of inflammatory diseases, which include gingivitis and periodontitis. The symptoms of gingivitis include gingival redness, swelling and bleeding. Gingivitis, if left untreated, can progress to periodontitis, leading to the destruction of periodontal ligaments, loss of alveolar bone and tooth loss [7,8]. The association between type 2 diabetes and oral health has been demonstrated conclusively. Hyperglycemia and poor oral health accelerate each other and constitute a vicious cycle, even in patients with prediabetes [9]. Additionally, improving oral health seems to improve glycemic control and vice versa—in both worsening and improving scenarios [6,9,10,11]. However, studies on the relationship between periodontal diseases and GDM have reported inconsistent results owing to heterogeneity in confounding factors and the diagnostic criteria for both periodontitis and GDM [12,13].

Eruption and extraction of the third molar are common in people aged 20–30 years [14]. A visible third molar seems to be a significant indicator of periodontal pathology progression during pregnancy [15]. Periodontitis, in turn, has been shown to be an independent risk factor for some pregnancy-related complications, including preterm delivery [16]. Still, there is no consensus as to whether women planning a pregnancy could benefit from extraction of the third molar. According to our knowledge, there are no studies evaluating the association of GDM or other types of diabetes with third molar symptoms.

The purpose of this study was to assess whether GDM is associated with an increase in oral symptoms, including third molar symptoms, and a greater need for oral health care by comparing self-reported oral health and oral symptoms in women with GDM to those in non-GDM pregnant women. The effect of the GDM stage—defined as recurrent and first GDM and pharmacologically treated and diet-treated GDM—on oral health symptoms was also studied. Women with GDM might have more oral health-related challenges that are not directly related to hyperglycemia. Therefore, the outcomes were adjusted for several lifestyle characteristics, as well as socioeconomic and clinical background factors.

## 2. Materials and Methods

The present multicenter case–control study is based on the Finnish Gestational Diabetes (FinnGeDi) study described previously [17]. In brief, 1146 women with GDM were recruited from delivery units in seven Finnish delivery hospitals, and the next consenting non-GDM mother (*n* = 1066) giving birth in the same hospital was recruited as a control between February 2009 and December 2012. Women with pre-pregnancy diabetes mellitus or multiple pregnancies were excluded from the study. The study participants completed a detailed questionnaire that included information about their lifestyle habits, as well as medical and family histories, comprising 1030 of 1146 (89.9%) women with GDM and 935 of 1066 (87.7%) controls. Subgroup analyses were performed separately. Clinical and register-based delivery data were obtained from hospital and maternity welfare clinic records and from the Medical Birth Register at the Finnish Institute for Health and Welfare.

The current study included 1030 women with GDM and 935 controls. Women with GDM who received insulin (*n* = 171, 8.7%) and/or metformin (*n* = 17, 0.9%) or both (*n* = 8, 0.4%) along with diet treatment were classified as pharmacologically treated women with GDM (*n* = 196, 19.0%), while those who received only diet treatment were classified as diet-treated women with GDM (*n* = 805, 78.2%). The treatment was unknown in 29 (2.8%) women with GDM, and they were excluded from the subgroup analyses. Furthermore, 797 (77.4%) of the women were diagnosed with GDM for the first time, and 233 (22.6%) had recurrent GDM.

A comprehensive screening policy for GDM based on the Finnish Current Care Guidelines was used—according to which a 2 h 75 g oral glucose tolerance test was performed in all women at 24th–28th week of gestation, except those at very low risk for GDM (primiparous women under 25 years with body mass index (BMI) < 25 kg/m^2^ and multiparous women under 40 years with BMI < 25 kg/m^2^ and without previous macrosomic births). In high-risk women (prior GDM, pre-pregnancy BMI > 35 kg/m^2^, glucosuria in early pregnancy, type 2 diabetes in a first-degree relative, oral glucocorticoid treatment or polycystic ovary syndrome), the oral glucose tolerance test was performed at the 12th–16th week of gestation and, if normal, repeated at 24–28 weeks. The cut-off values for the glucose concentrations were set according to the recommendation in the Finnish Current Care Guidelines: fasting ≥ 5.3 mmol/L, 1 h ≥ 10.0 mmol/L and 2 h ≥ 8.6 mmol/L. In accordance with the Finnish Current Care Guidelines, all women received diet counseling, and in cases where target levels (fasting capillary blood glucose < 5.5 mmol/L and 1 h postprandial glucose < 7.8 mmol/L) could not be reached with diet alone, pharmacological treatment (insulin or metformin) was started [1].

A free oral health care assessment (through an interview or other means determined by an oral health care professional) is recommended for all primiparous women and their spouses, as well as multiparous women with oral symptoms in primary health care centers, in accordance with Finnish national guidelines [18,19].

Oral health was assessed based on a questionnaire answered by most of the women within one week before or after delivery in gestational weeks from 28.1 to 42.6 (median 40.0, interquartile range 39.0–40.9). Questions concerned the need for oral care, removed third molar, symptoms of the third molar, gingival bleeding and restored teeth, as detailed in Appendix A, with the same questionnaire as other background characteristics. The need for oral care was self-assessed and dichotomized into “high or intermediate need” and “low, very low or no need”. Equally, the question regarding gingival bleeding was dichotomized into “weekly or more often” and “rarely”. Questions regarding the removal of third molars were dichotomized into “yes” or “no”. Regarding restored teeth, “none”, “1–4” and “5–10” were combined as “0–10 restored teeth” and compared to “over 10 restored teeth”.

Maternal age at delivery and parity were obtained from hospital records. Pre-pregnancy BMI was calculated from self-reported height and weight during the first antenatal visit, which is typically in the first trimester. Weight gain was calculated as the difference between self-reported pre-pregnancy weight and the last measured weight at the maternity clinic after 35 weeks of gestation. Data on blood pressure (BP) were obtained from hospital and maternity clinic records. Chronic hypertension was defined as a systolic BP over 140 mmHg and/or diastolic BP over 90 mmHg measured at least twice or in cases where medication was prescribed for hypertension before 20 weeks of gestation. Gestational hypertension was defined as BP elevation observed after 20 weeks of gestation. Pre-eclampsia was defined as BP over 140/90 mmHg and proteinuria of at least 300 mg/day or as chronic hypertension and proteinuria. Based on the questionnaire data, educational attainment was classified as basic or less (≤9 years), secondary (10 to 12 years), lower-level tertiary (13 to 15 years) and upper-level tertiary (over 15 years). Smoking during pregnancy was scaled as “yes” or “no” from the questionnaires or the Medical Birth Register. Data on asthma and insomnia and/or mental disorders, which are risk factors for periodontal diseases [7], were obtained from the questionnaires and the medical records. Self-reported hyperemesis was scaled with a visual analogic scale of 0 (no hyperemesis) to 10 (the worst possible hyperemesis).

Statistical analyses were performed using SPSS 25.0 software (IBM). Differences between the control and GDM groups were analyzed with Student’s *t*-test for normally distributed continuous parameters and with the Mann−Whitney U test in the case of skewed distribution. Categorical data were analyzed using Pearson’s chi-square test or Fisher’s exact test in the case of a small sample size. Continuous data are presented as means and standard deviations or medians and interquartile ranges. Categorical data are reported as numbers (percentages).

Multivariable logistic regression analyses were used to estimate odds ratios (ORs) and 95% confidence intervals (CIs). Maternal age, parity and maternal BMI are potential confounding factors that are generally known to have an effect on both GDM risk and oral health. Asthma, especially the medication used, insomnia and/or mental disorders and smoking potentially have an effect on oral health, but their relation to GDM is unclear or might be indirect through other confounders, e.g., education. These variables were chosen to analyze the effects of different potential confounders on oral health outcomes: Model 1 included maternal age and parity, Model 2 included Model 1 + pre-pregnancy BMI and Model 3 included Model 2 + smoking during pregnancy, education, history of asthma and insomnia and/or mental disorders. Additionally, chronic hypertension, gestational hypertension and pre-eclampsia were included in Model 3 when gingival bleeding was analyzed since hypertensive disorders may have an effect on gingival bleeding [20]. Hyperemesis was included in Model 3 when analyzing the number of restored teeth, as vomiting may have an erosive effect on teeth [21]. The directed acyclic graph summarizing the hypothetical causality between GDM, oral health and potential confounding variables used in the logistic regression analyses is shown in Figure 1.

Women with missing oral health answers (2.7%) or background data (1.0%) were excluded from the particular analyses (total: 1.3% missing values). To study whether missing values have an effect on results between different models, we performed additional multivariable regression analyses excluding all cases with missing variables in Model 3 from the analyses. A value of *p* < 0.05 was considered statistically significant.

Regarding the statistical power of our study, a power of 0.80 and a significance level of 0.50 would be able to detect a 20% difference in the incidences of oral health outcomes between women with GDM and the controls.

The study was approved by the Ethics Committee of the Northern Ostrobothnia Hospital District (reference number 33/2008) and conformed to the European Medicines Agency guidelines for good clinical practice and the Declaration of Helsinki. Each participant gave written informed consent.

## 3. Results

### 3.1. Clinical Characteristics

Table 1 presents the clinical characteristics of the study participants. Women with GDM were older, and their parity and pre-pregnancy BMIs were higher compared to the controls. Women with GDM suffered more often from asthma and insomnia and/or mental disorders, as well as chronic and gestational hypertension, and their educational attainment was lower than that of the controls. Women with GDM also delivered at earlier gestational weeks, and their cesarean section and induction-of-labor rates were higher than those of the controls. Smoking during pregnancy did not differ between women with GDM and controls. Women with GDM were further divided into subgroups: pharmacologically treated GDM (diet- and insulin- and/or metformin-treated GDM), diet-treated GDM, recurrent GDM (GDM in previous pregnancy/pregnancies) and first-onset GDM (GDM first diagnosed in current pregnancy) (Table 2 and Table 3).

### 3.2. Oral Health

Table 4 presents the oral health questions and the proportions of answers in the whole study population and in the subgroups of women with GDM. Of all of the women with GDM, 31.1% reported a high or intermediate need for oral health care compared to 24.5% of the controls (OR 1.39; 95% CI 1.13–1.69), and this difference was observed in all models (Table 4, Figure 2). Similarly, women with pharmacologically treated GDM had an increased need for oral health care compared to the controls (32.5% vs. 24.5%; OR 1.48; 95% CI 1.06–2.07), with the difference observed in all models (Table 4, Figure 2). However, the difference between pharmacologically treated and diet-treated GDM was not significant (OR 1.08; 95% CI 0.77–1.52) (Figure 2). In addition, 38.1% of the women with recurrent GDM reported a high or intermediate need for oral health care (OR 1.90; 95% CI 1.40–2.58) compared to controls in all models (Table 4, Figure 2). A higher need for oral care was also observed when comparing women with recurrent GDM to those with first-onset GDM in Models 1 and 2 (29.0%; Model 2 OR 1.41; 95% CI 1.01–1.97) but not in Model 3 (OR 1.31; 95% CI 0.93–1.86) (Figure 2).

Women with GDM had their third molars removed more often than the controls (OR 1.27; 95% CI 1.05–1.69; 2.0, 1.6 (mean, standard deviation) vs. 1.8, 1.6; *p* < 0.05) (Table 4, Figure 3), but parity, maternal age (Model 1 OR 0.95; 95% CI 0.78–1.16) and other clinical background risk factors largely explained the difference (Figure 3). Women with pharmacologically treated GDM reported symptoms of the third molar more often than the controls (46.9% vs. 36.1%; OR 1.57; 95% CI 1.15–2.15) or women with diet-treated GDM (38.0%, Model 3 OR 1.50; 95% CI 1.07–2.11) (Table 4, Figure 4).

Women with GDM had more restored teeth (over 10 vs. 0–10: 11.7% vs. 8.4%; OR 1.44; 95% CI 1.06–1.94) than controls, but maternal age and other background risk factors largely explained the difference. Similarly, women with recurrent GDM had an increased number of restored teeth (15.2% vs. 8.4%; OR 2.00; 95% CI 1.30–3.08) compared to the controls (Table 4, Figure 5). There were no significant differences in self-reported gingival bleeding in women with GDM or its subgroups compared to the controls (Table 4).

### 3.3. Oral Health and Background Risk Factors

The associations between oral health and maternal background factors are shown in Appendix A. An increased need for oral health care was associated with a higher number of restored teeth, a higher frequency of gingival bleeding and a higher number of removed third molars. Furthermore, an increased need for oral health care was also associated with several background risk factors, including younger maternal age, higher parity, higher pre-pregnancy BMI, maternal smoking, lower educational attainment, and higher frequency of insomnia and/or mental disorders.

Women who had more symptoms of the third molar had a higher number of removed third molars but did not report an increased need for oral health care. Smoking, lower educational attainment, asthma, and insomnia and/or mental disorders were associated with increased symptoms of the third molar. Gingival bleeding and a removed third molar were associated with a higher frequency of restored teeth.

Frequent gingival bleeding was associated with lower educational attainment, younger maternal age, and insomnia and/or mental disorders. A higher number of removed third molars and restored teeth, in turn, were associated with a higher maternal age, BMI and parity.

Additional analyses using cases without any missing variables did not significantly change the results (Appendix A).

## 4. Discussion

The present study showed that the need for oral health care is higher among women with GDM compared to non-GDM pregnant women, even after adjusting for several maternal characteristics. As a novel finding, we found that self-reported symptoms of the third molar are more common in women with pharmacologically treated GDM compared to controls or diet-treated GDM.

Conflicting results have been reported in meta-analyses studying the association between periodontitis and GDM, with one study reporting an association (OR 1.66; 95% CI 1.17–2.36) [12], while another was inconclusive because of the heterogeneity of the included studies [13]. Because the questionnaires in the present study were not specific to periodontitis and a clinical examination was not performed, our findings of an increased need for oral health care are not directly comparable with those of previous studies, in which periodontitis was confirmed by clinical examination [7]. However, the increased need for oral health care in women with GDM seemed to reflect overall oral symptoms, as it was associated with a higher number of restored teeth. Similar observations were reported in a Danish survey in which pregnant women who experienced symptoms of gingivitis reported poor oral health [23]. Higher BMI, but not maternal age, was associated with an increased need for oral health care in women with GDM in our study. However, the increased need for oral health care in women with GDM, especially in women with recurrent GDM and pharmacologically treated GDM, could not be explained by a higher BMI, maternal age or any other background factor when compared to controls. The exposure time to hyperglycemia in GDM is relatively short and unlikely to cause oral problems. In the case of recurrent GDM, exposure is longer, and in the case of pharmacologically treated GDM, hyperglycemia is more severe. These conditions could more likely be accompanied by undiagnosed type 2 diabetes mellitus or impaired glucose regulation between pregnancies, which might explain our findings. However, when comparing women with recurrent GDM to women with first GDM, higher BMI and other background risk factors seem to explain the difference in the need for oral care. Studies have reported inconsistent results regarding gingival bleeding in GDM, with studies reporting no difference and increased gingival bleeding in women with GDM [24,25]. We observed that GDM is not associated with increased gingival bleeding. It has been previously shown that the risk of gingivitis is increased in nonpregnant young women with a higher BMI, regardless of glycemic levels [26,27]. Hormonal changes and smoking during pregnancy may affect gingival bleeding without other pathological mechanisms [7,28]. In summary, our study indicates that GDM, especially pharmacologically treated and recurrent GDM, is associated with an increased need for oral health care, probably due to oral symptoms, but the role of periodontitis remains unclear. Thus, it might be speculated that impaired glycemic condition may reflect increased oral symptoms that necessitate an increased need for oral health care, but the causal relationship remains uncertain.

In general, third molar symptoms, including pain and discomfort, indicate eruption, inflammation and/or infection [29]. In our study, women with pharmacologically treated GDM suffered more often from third molar symptoms than women with diet-treated GDM or the controls, and these symptoms could not be explained by obesity, parity, maternal age or other maternal background factors. In addition, women with a history of third molar removal had symptoms of the third molar during pregnancy. This may reflect a situation in which some third molars were removed prior to pregnancy, probably due to symptomatic reasons. A partly erupted third molar seems to be a significant indicator of periodontal disease progression not only during pregnancy but also in nonpregnant healthy women [15,29]. Therefore, it is recommended that partly erupted and impacted third molars be extracted by the age of 25 years [29]. To the best of our knowledge, the role of hyperglycemia during pregnancy in the pathology of the third molar has not been previously studied. This phenomenon could be explained by hyperglycemia or by the effect of the medication used for GDM, or there might be other underlying factors that we were not able to take into account in this study setting. There is no evidence in the current literature that hyperglycemia is specifically associated with third molar symptoms in type 1 or type 2 diabetes. Pharmacological treatment may affect inflammation mechanisms, and metformin might have an anti-inflammatory effect [30]. However, only 25 (1.3%) participants received metformin treatment. As a conclusion, we are not able to explain our novel finding that women with pharmacologically treated GDM report symptoms of the third molars almost 1.7-fold more often than the controls or 1.5-fold more often than diet-treated GDM. This topic needs further investigation to determine whether women at risk of developing severe forms of GDM require more preventive oral health care and more attention on their third molar status.

Studies have reported that poor oral health and type 2 diabetes could have bidirectional causality [10,12,13]. People with type 2 diabetes have poorer oral health [31], and poor oral health further impairs glycemic control [10]. This may expose women to systemic low-grade inflammation and its consequences, including worsening hyperglycemia during and after pregnancy. Our study indicates that overall oral health is poorer in women with GDM than in non-GDM women. Furthermore, recurrent GDM and GDM requiring pharmacological treatment seem to have a stronger association with poor oral health outcomes independent of maternal age, suggesting that severe hyperglycemia is followed by certain oral health problems. Poor oral health is generally associated with higher parity, obesity, smoking and lower educational attainment, all of which are risk factors for GDM. Obesity, both with and without type 2 diabetes, has been shown to be associated with periodontal diseases and chronic systemic low-grade inflammation [10,26,27]. Our study demonstrates an association between GDM and poor oral health, both of which are affected by several socioeconomic and lifestyle-related factors. In our study, lower socioeconomic attainment and an increased BMI were associated with GDM and poorer oral health in the whole study population, with the exception of symptoms of the third molar. Nevertheless, when adjusting for several health, lifestyle-related and socioeconomic factors, women with GDM reported an increased need for oral care, and pharmacologically treated women with GDM more often reported symptoms of the third molar compared to controls. Advanced maternal age, a risk factor for GDM, was found to be associated with a higher number of restored teeth and removed third molars but not with third molar symptoms or gingival bleeding. In addition, after adjustments for advanced age and parity, the association of GDM with the number of restored teeth or removal of third molars did not remain significant. No further remarkable changes in ORs after adjustments for BMI or other maternal background factors were observed. Therefore, advanced age seems to explain these outcomes instead of GDM or other background risk factors. We found that younger women tended to report a higher need for oral health care and frequent cases of gingival bleeding, suggesting insufficient availability and use of oral health care in these women. In Finland, enhanced public preventive oral health care was active from the late 1970s to the early 1990s. The prevention program, supported by the Ottawa Charter for Health Promotion (1986) [32], was scaled back in the 1990s due to the economic recession. The total burden of low-grade inflammatory processes, including hyperglycemia, oral pathology and obesity, could pose a serious overall health risk manifested as increased oral health problems. These problems seem to compound in women who already lead an unhealthy lifestyle and are socioeconomically more vulnerable.

To the best of our knowledge, there are no published clinical trials studying the effects of intensified oral health care during pregnancy or oral health outcomes in pregnant women with GDM. However, the pregnancy period seems to be a favorable time to optimize maternal health behavior because pregnant women have been found to be responsive to health advice [33]. A five-year Finnish follow-up study found that high-risk women with GDM had better oral health than high-risk controls after lifestyle intervention before or during pregnancy [34]. The effect of intensified lifestyle intervention on oral health outcomes was not reported. However, in usual clinical practice, women with GDM receive more intensive lifestyle and diet counseling than women without GDM. This may beneficially affect later oral health outcomes. Even though there is no evidence that intensified oral health care can prevent the onset of type 2 diabetes after GDM, it could prevent periodontal disorders in this high-risk group.

Our study has several strengths and limitations. The evaluation of oral health was fundamentally based on self-reported data and may have included typical biases related to recall and interpretation of the questions. The response rate to the oral health questionnaire was high (88.8%), with only a few blank answers (0.4%) or undetermined responses (I cannot say (2.8%)), reflecting that overall recall bias seems to be low. However, in the question concerning the number of restored teeth, the proportion of blank or undetermined responses was relatively high (7.3%), indicating a potential recall bias in this specific question. On the other hand, excluding all participants with any missing values did not substantially change the results (Appendix A). The question concerning a need for oral care could have been interpreted either as an experience of oral symptoms or a need for medical treatment, as it was not specified in the questionnaire. This question most likely reflects both symptoms and the overall need for medical treatment, because we found that an increased need for oral care was associated with a higher number of restored teeth, a higher frequency of gingival bleeding and a higher number of removed third molars. The time period of oral health outcomes, for example, removal of the third molars or symptoms of the third molars, was not specified in the questionnaire, which limits the estimation of whether these outcomes occurred during or before pregnancy. The detection and misclassification of oral health outcomes could be considered potential limitations, especially when estimating the existence of caries (a stage of gingivitis) or its progression into periodontitis, as the diagnosis of periodontitis should be based on an oral examination.

The strengths of this study include a well-defined, relatively large case–control cohort with detailed information on medical history, lifestyle and perinatal data. The analyses were adjusted for several background factors in relation to both GDM and oral health. Furthermore, by combining register and questionnaire data, recall and social desirability bias were reduced. In Finland, GDM diagnosis is well defined, and antenatal management of GDM is nationally homogeneous due to national guidelines [1], with attendance at public maternity clinics being 99.7% [19,35].

## 5. Conclusions

In this study, women with GDM reported an increased need for oral health care compared to non-GDM pregnant women. Furthermore, an increased need for oral care was more common in women with pharmacologically treated or recurrent GDM. As a novel finding, this study shows that third molar symptoms were more common among women with pharmacologically treated GDM than in controls or in diet-treated GDM. GDM is a prediabetic state, especially in the case when pharmacological treatment is required during pregnancy or GDM recurs. These findings raise a novel discussion of the possible relation between GDM and oral health. An experimental setting should be established to study whether this high-risk group for developing type 2 diabetes in the future would benefit from more intensive oral care and oral health counseling.

## Figures and Tables

**Figure 1 ijerph-19-10711-f001:**
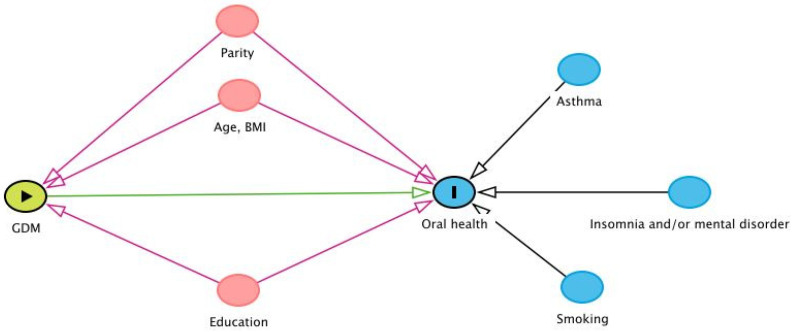
Directed acyclic graph showing paths between gestational diabetes and oral health. GDM, gestational diabetes mellitus; BMI, body mass index. The green oval represents exposure, the blue (I) oval is an outcome, the pink ovals are the precursors of exposure and the outcome (confounders), the clear blue ovals are the ancestors of the outcome (potential confounders), the green arrow demonstrates the hypothetical causal path, and the pink arrows demonstrate biasing paths.

**Figure 2 ijerph-19-10711-f002:**
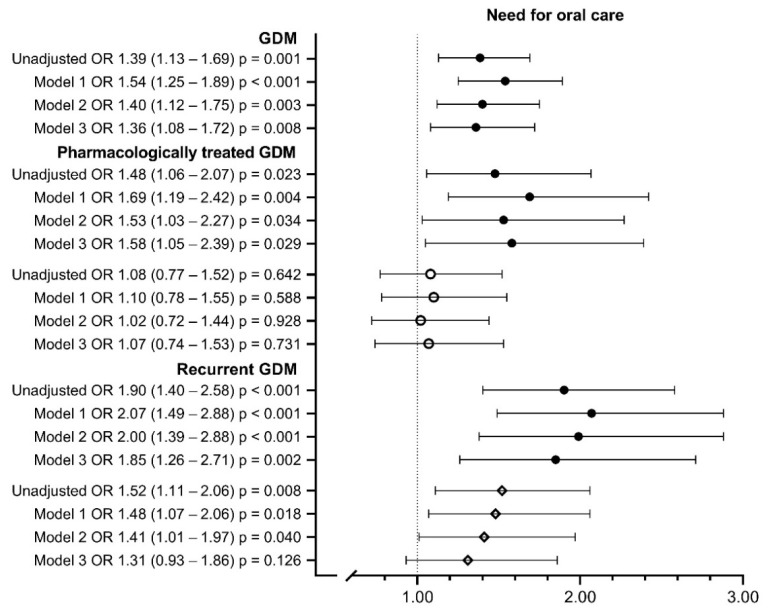
Need for oral care in women with gestational diabetes and its subgroups compared to their controls. Model 1 includes maternal age at delivery and parity; Model 2 includes Model 1 + pre-pregnancy BMI; Model 3 includes Model 2 + smoking during pregnancy, educational attainment, asthma and insomnia and/or mental disorders. ● *Compared to controls*. *◦ Compared to women with diet-treated GDM*. *◊ Compared to women with first-onset GDM*. *Missing values were* 0.8–1.0% *in Models* 1 *and* 2 *and* 5.7–7.0% *in Model* 3. *OR, odds ratio; CI, confidence interval*.

**Figure 3 ijerph-19-10711-f003:**
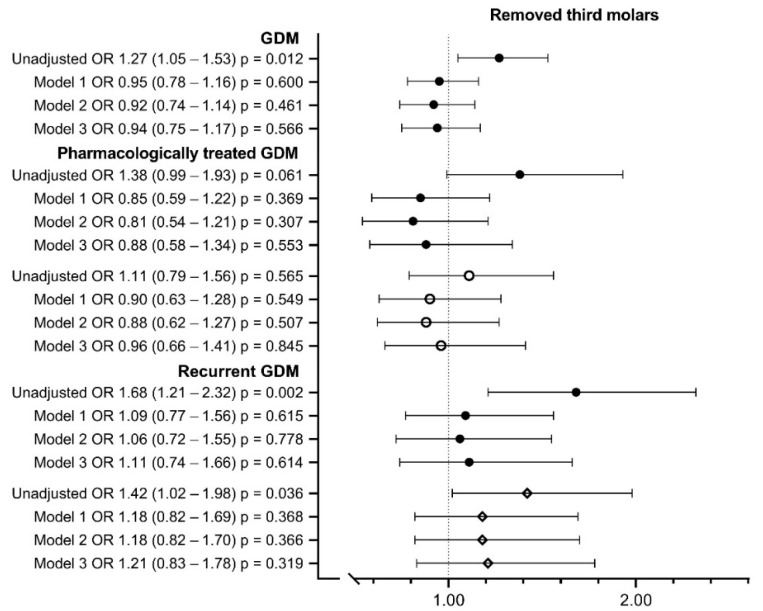
Removal of third molar (yes/no) in women with gestational diabetes. Model 1 includes maternal age at delivery and parity; Model 2 includes Model 1 + pre-pregnancy BMI; Model 3 includes Model 2 + smoking during pregnancy, educational attainment, asthma and insomnia and/or mental disorders. *● Compared to controls. ◦ Compared to women with diet-treated GDM. ◊ Compared to women with first-onset GDM. Missing values were* 1.1–1.4% *in Models* 1 *and* 2 *and* 5.7–7.3% *in Model* 3. *OR, odds ratio; CI, confidence interval*.

**Figure 4 ijerph-19-10711-f004:**
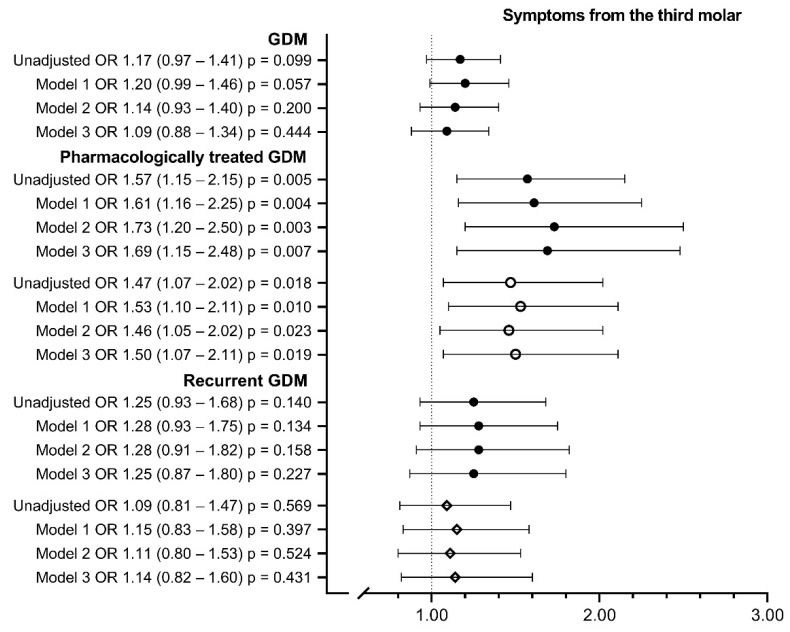
Third molar symptoms in women with gestational diabetes and its subgroups. Model 1 includes maternal age at birth and parity; model 2 includes model 1 + pre-pregnancy BMI; model 3 includes model 2 + smoking during pregnancy, educational attainment, asthma and insomnia and/or mental disorders. *● Compared to controls.*
*◦*
*Compared to women with diet-treated GDM. ◊ Compared to women with first-onset GDM. Missing values were* 2.3–3.4% *in Models* 1 *and* 2 *and* 7.7–8.1% *in Model* 3. *OR, odds ratio; CI, confidence interval*.

**Figure 5 ijerph-19-10711-f005:**
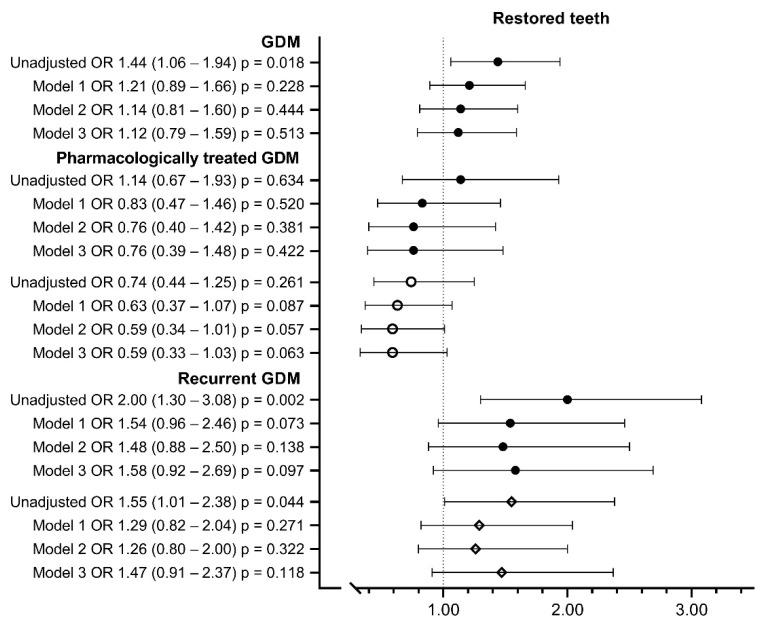
Number of restored teeth in women with gestational diabetes. Model 1 includes maternal age at birth and parity; Model 2 includes Model 1 + pre-pregnancy BMI; Model 3 includes Model 2 + smoking during pregnancy, educational attainment, asthma, insomnia and/or mental disorder and hyperemesis. *● Compared to controls. ◦ Compared to women with diet-treated GDM. ◊ Compared to women with first-onset GDM. Missing values were* 6.5–8.1% *in Models* 1 *and* 2 *and* 12.2–13.2% *in Model* 3. *OR, odds ratio; CI, confidence interval*.

**Table 1 ijerph-19-10711-t001:** Maternal and perinatal characteristics of the participants (*n* = 1965).

Characteristic	Controls	*n*	GDM	*n*	*p* Value
Age at delivery (y) (mean, SD)	29.4 (5.0)	935	32.0 (5.3)	1030	<0.001
Pre-pregnancy BMI (kg/m^2^) (median, IQR)	22.8 (20.8–25.6)	935	26.0 (23.8–31.6)	1029	<0.001
Weight gain during pregnancy (kg) (mean, SD)	14.9 (5.0)	908	12.4 (5.7)	948	<0.001
Parity (median, IQR)	0 (0–1)	935	1 (0–2)	1030	<0.05
Primiparous	475 (50.8%)	935	437 (42.4%)	1030	<0.001
Early-onset GDM ^a^	-	-	295 (28.6%)	976	
Smoking during pregnancy	143 (15.3%)	935	166 (16.1%)	1030	>0.05 ^F^
Education		935		1030	<0.05 ^F^
Basic or less	42 (4.5%)		68 (6.6%)		
Secondary	426 (45.6%)		486 (47.2%)		
Lower-level tertiary	231 (24.7%)		270 (26.2%)		
Upper-level tertiary	236 (25.2%)		206 (20.0%)		
Asthma	77 (8.6%)	900	112 (11.4%)	981	<0.05 ^F^
Insomnia and/or mental disorders	102 (11.3%)	905	142 (14.5%)	977	<0.05 ^F^
Chronic hypertension ^b^	47 (5.0%)	935	168 (16.3%)	1029	<0.001 ^F^
Gestational hypertension ^c^ and/or pre-eclampsia ^d^	177 (16.6%)	935	304 (26.6%)	1029	<0.001 ^F^
Gestational age at delivery (weeks) (median, IQR)	40.3 (39.4–41.1)	935	39.7 (38.7–40.6)	1030	<0.001
Induction of labor	342 (32.1%)	935	515 (44.9%)	1030	<0.001 ^F^
Cesarean section	116 (12.4%)	935	212 (20.6%)	1030	<0.001 ^F^
Mean birth weight (SD) (g) (mean, SD)	3600 (496)	935	3700 (507)	1030	<0.05
Birth weight SD score ^e^ (mean, SD)	−0.10 (0.98)	935	0.25 (1.11)	1030	<0.001
LGA ^e^	28 (2.6%)	935	64 (5.6%)	1030	<0.001 ^F^

Data shown as mean (standard deviation), median (interquartile range) or number (percentage). *n* denotes the number of subjects. The number of subjects varies owing to a lack of data for some parameters. ^a^ GDM diagnosed before 20 weeks of gestation. ^b^ Systolic blood pressure over 140 mmHg and/or diastolic over 90 mmHg at least twice before 20 weeks of gestation. ^c^ Systolic blood pressure over 140 mmHg and/or diastolic over 90 mmHg at least twice after 20 weeks of gestation. ^d^ Blood pressure of 140/90 mmHg and proteinuria > 300 mg/day, or chronic hypertension and proteinuria. ^e^ Birth weight SDs and LGA > +2 SDs defined by Sankilampi et al.’s [22] criteria. ^F^ Fisher’s exact test. Categorical variables were determined by crosstabs and Pearson’s chi-square test and, where applicable, by Fisher’s exact test, and parametric values were analyzed by independent samples *t*-test. Continuous variables were analyzed with Student’s *t*-test and with the Mann−Whitney U test in the case of skewed distribution. GDM, gestational diabetes mellitus; BMI, body mass index; SD, standard deviation; LGA, large for gestational age; IQR, interquartile range.

**Table 2 ijerph-19-10711-t002:** Characteristics of the controls and women with pharmacologically treated and diet-treated gestational diabetes.

Characteristic	Controls	*n*	Pharmacologically Treated GDM ^f^	*n*	Diet-Treated GDM	*n*
Age at delivery (y) (mean, SD)	29.4 (5.0)	935	33.7 (5.5) *^/‡^	196	31.7 (5.3) *	805
Pre-pregnancy BMI (kg/m^2^) (median, IQR)	22.8 (20.8–25.6)	935	29.0 (24.6–34.4) *^/‡^	196	26.6 (23.6–30.9) *	804
Weight gain during pregnancy (kg) (mean, SD)	14.9 (5.0)	908	10.7 (6.3) *^/‡^	167	12.7 (5.5) *	756
Parity (median, IQR)	0 (0–1)	935	1 (0–2) *^/§^	196	0 (0–2)	805
Primiparous	475 (50.8%)	935	65 (33.2%) *^/§/F/F^	196	362 (45.0%) ^†/F^	805
Early-onset GDM ^a^	-	-	119 (58.9%) ^‡/F^	177	202 (23.5%)	776
Smoking during pregnancy	143 (15.3%)	935	26 (13.3%) ^F/F^	196	136 (16.9%)	805
Education		935		196	^F^	805
Basic or less	42 (4.5%)		13 (6.6%)		51 (6.3%)	
Secondary	426 (45.6%)		108 (55.1%)		363 (45.1%)	
Lower-level tertiary	231 (24.7%)		42 (21.4%)		221 (27.5%)	
Upper-level tertiary	236 (25.2%)		33 (16.8%)		170 (21.1%)	
Asthma	77 (8.6%)	900	23 (12.2%)	189	86 (11.2%)	766
Insomnia and/or mental disorders	102 (11.3%)	905	37 (19.5%) ^†/§^	190	103 (13.5%)	761
Chronic hypertension ^b^	47 (5.0%)	935	36 (18.4%) *	196	123 (15.3%) *^/F^	804
Gestational hypertension ^c^ and/or pre-eclampsia ^d^	177 (16.6%)	935	61 (27.6%) *	196	234 (26.2%) *^/F^	804
Gestational age at delivery (weeks) (median, IQR)	40.3 (39.4−41.1)	935	39.1 (38.3–39.8) *^/‡^	196	39.9 (39.0−40.7) *	805
Induction of labor	342 (32.1%)	935	143 (64.7%) *^/‡/F/F^	196	355 (39.6%) *	805
Cesarean section	116 (12.4%)	935	51 (26.0%) *^/‡^	196	152 (18.9%) *^/F^	805
Mean birth weight (SD) (g) (mean, SD)	3600 (496)	935	3700 (494) ^†^	196	3600 (501) ^†^	805
Birth weight SD score ^e^ (mean, SD)	−0.10 (0.98)	935	0.53 (1.30) *^/‡^	196	0.19 (1.03) *	805
LGA ^e^	28 (2.6%)	935	25 (11.3%) *^/‡^	196	36 (4.0%) ^F^	805

Data shown as mean (standard deviation), median (interquartile range) or number (percentage). *n* denotes the number of subjects. The number of subjects varies owing to a lack of data for some parameters. ^a^ GDM diagnosed before 20 weeks of gestation. ^b^ Systolic blood pressure over 140 mmHg and/or diastolic over 90 mmHg at least twice before 20 weeks of gestation. ^c^ Systolic blood pressure over 140 mmHg and/or diastolic over 90 mmHg at least twice after 20 weeks of gestation. ^d^ Blood pressure of 140/90 mmHg and proteinuria >300 mg/day, or chronic hypertension and proteinuria. ^e^ Birth weight SDs and LGA > +2 SDs defined by Sankilampi et al.’s [22] criteria. ^f^ Diet- and insulin- and/or metformin-treated GDM. ^F^ Fisher’s exact test. Women with pharmacologically treated GDM compared to women with diet-treated GDM or women with recurrent GDM compared to women with first-onset GDM. * *p* < 0.001 compared to controls. ^†^ *p* < 0.05 compared to controls. ^‡^ *p* < 0.001 compared to women with diet-treated GDM. ^§^ *p* < 0.05 compared to women with diet-treated GDM. Categorical variables were determined by crosstabs and Pearson’s chi-square test and, where applicable, by Fisher’s exact test, and parametric values were analyzed by independent samples *t*-test. Continuous variables were analyzed with Student’s *t*-test and with the Mann−Whitney U test in the case of skewed distribution.

**Table 3 ijerph-19-10711-t003:** Characteristics of the controls and women with recurrent or first-onset gestational diabetes.

Characteristic	Controls	*n*	Recurrent GDM ^f^	*n*	First-Onset GDM ^g^	*n*
Age at delivery (years) (mean, SD)	29.4 (5.0)	935	33.8 (5.5) *^/‡^	233	31.5 (5.2) *	797
Pre-pregnancy BMI (kg/m^2^) (median, IQR)	22.8 (20.8–25.6)	935	28.4 (24.8−33.2) *^/‡^	233	26.6 (23.5−31.0) *	796
Weight gain during pregnancy (kg) (mean, SD)	14.9 (5.0)	908	11.2 (5.7) *^/^^§^	204	12.7 (5.7) *	744
Parity (median, IQR)	0 (0−1)	935	2 (1−3) *^/‡^	233	0 (0−1) ^†^	797
Primiparous	475 (50.8%)	935	1 (0.4%) *^/‡/F/F^	233	436 (54.7%)	797
Early-onset GDM ^a^	-		141 (60.5%) ^‡/F^	211	154 (19.3%)	765
Smoking during pregnancy	143 (15.3%)	935	30 (12.9%) ^F/F^	233	136 (17.1%) ^F^	797
Education		935		233	^F^	797
Basic or less	42 (4.5%)		17 (7.3%)		51 (6.4%)	
Secondary	426 (45.6%)		129 (55.4%)		357 (44.8%)	
Lower-level tertiary	231 (24.7%)		51 (21.9%)		219 (27.5%)	
Upper-level tertiary	236 (25.2%)		36 (15.5%)		170 (21.3%)	
Asthma	77 (8.6%)	900	22 (10.0%) ^F^	220	90 (11.8%) ^†^	761
Insomnia and/or mental disorders	102 (11.3%)	905	31 (14.0%) ^F^	221	111 (14.7%) ^†^	756
Chronic hypertension ^b^	47 (5.0%)	935	40 (17.2%) *	232	128 (16.1%) *^/F^	797
Gestational hypertension ^c^ and/or pre-eclampsia ^d^	177 (16.6%)	935	56 (22.1%) ^†/F^	232	224 (28.1%) *^/F^	797
Gestational age at delivery (weeks) (median, IQR)	40.3 (39.4−41.1)	935	39.3 (38.3−40.1) *^/‡^	233	39.7 (38.9−40.6) *	797
Induction of labor	342 (32.1%)	935	137 (53.9%) *^/‡^	233	331 (41.5%) *	797
Cesarean section	116 (12.4%)	935	29 (12.4%) ^‡/F^	233	183 (23.0%) *^/F^	797
Mean birth weight (SD) (g) (mean, SD)	3600 (496)	935	3700 (485) *^/^^§^	233	3700 (503) ^†^	797
Birth weight SD score ^e^ (mean, SD)	−0.10 (0.98)	935	0.39 (1.11) *	233	0.23 (1.12) *	797
LGA ^e^	28 (2.6%)	935	21 (8.3%) *	233	40 (5.0%) ^†/F^	797

Data shown as mean (standard deviation), median (interquartile range) or number (percentage). *n* denotes the number of subjects. **^a^** GDM diagnosed before 20 weeks of gestation. **^b^** Systolic blood pressure over 140 mmHg and/or diastolic over 90 mmHg at least twice before 20 weeks of gestation. **^c^** Systolic blood pressure over 140 mmHg and/or diastolic over 90 mmHg at least twice after 20 weeks of gestation. ^d^ Blood pressure of 140/90 mmHg and proteinuria > 300 mg/day, or chronic hypertension and proteinuria. **^e^** Birth weight SDs and LGA > +2 SDs defined by Sankilampi et al.’s [22] criteria. ^f^ GDM diagnosed in previous pregnancy/pregnancies. ^g^ GDM diagnosed for the first time during current pregnancy. ^F^ Fisher’s exact test. Women with pharmacologically treated GDM compared to women with diet-treated GDM or women with recurrent GDM compared to women with first-onset GDM. ***** *p* < 0.001 compared to controls. **^†^** *p* < 0.05 compared to controls. **^‡^** *p* < 0.001 compared to women with first-onset GDM. **^§^** *p* < 0.05 compared to women with first-onset GDM. Categorical variables were determined by crosstabs and Pearson’s chi-square test and, where applicable, by Fisher’s exact test, and parametric values were analyzed by independent samples *t*-test. Continuous variables were analyzed with Student’s *t*-test and with the Mann−Whitney U test in the case of skewed distribution.

**Table 4 ijerph-19-10711-t004:** Self-reported oral health of controls and women with gestational diabetes and its subgroups.

Parameters	Control (*n* = 935)	GDM (*n* = 1030)	Subgroups of GDM			
			Pharmacologically Treated GDM ^a^ (*n* = 196)	Diet-Treated GDM (*n* = 805)	Recurrent GDM ^b^ (*n* = 233)	First-Onset GDM ^c^ (*n* = 797)
	*n* (%)	*n* (%)	*n* (%)	*n* (%)	*n* (%)	*n* (%)
**Need for oral care**						
High or intermediate	229 (24.5%)	319 (31.1%) *^/F^	63 (32.5%) *	247 (30.7%) *	88 (38.1%) ^†/‡^	231 (29%) *
Low, very low or no	698 (74.7%)	702 (68.4%)	130 (67%)	552 (68.7%)	141 (61%)	561 (70.5%)
Cannot say	7 (0.7%)	6 (0.6%)	1 (0.5%)	5 (0.6%)	2 (0.9%)	4 (0.5%)
Total	934 (100%)	1027 (100%)	194 (100%)	804 (100%)	231 (100%)	796 (100%)
**Removed third molar**						
Mean (SD)	1.8 (1.6)	2.0 (1.6) *	2.1 (1.6) *	2.0 (1.6) *	2.3 (1.6) ^†/§^	1.9 (1.6)
Yes	581 (62.4%)	694 (67.6%) *	135 (68.9%) *^/F/F^	539 (67.2%) *^/F^	170 (73.3%) *^/‡/F/F^	524 (65.9%) ^F^
No	344 (36.9%)	323 (31.4%)	58 (29.6%)	256 (31.9%)	60 (25.9%)	263 (33.1%)
Cannot say	6 (0.6%)	10 (1%)	3 (1.5%)	7 (0.9%)	2 (0.9%)	8 (1%)
Total	931 (100%)	1027 (100%)	196 (100%)	802 (100%)	232 (100%)	795 (100%)
**Third molar symptoms**						
Yes	335 (36.1%)	410 (39.9%) ^F^	92 (46.9%) *^/‡^	305 (38%)	96 (41.2%)	314 (39.5%)
No	567 (61.2%)	594 (57.8%)	99 (50.5%)	482 (60.1%)	130 (55.8%)	464 (58.4%)
Cannot say	25 (2.7%)	23 (22.2%)	5 (2.6%)	15 (1.9%)	7 (3%)	16 (2%)
Total	925 (100%)	1027 (100%)	196 (100%)	802 (100%)	233 (100%)	794 (100%)
**Gingival bleeding**						
Weekly or more often	90 (9.6%)	107 (10.4%) ^F^	21 (10.7%)	81 (10.1%)	19 (8.2%) ^F/F^	88 (11.1%)
Rarely	837 (89.7%)	910 (88.8%)	175 (89.3%)	712 (89.0%)	212 (91.4%)	698 (88.0%)
Cannot say	6 (0.6%)	8 (0.8%)	0 (0)	7 (0.9%)	1 (0.4%)	7 (0.9%)
Total	933 (100%)	1025 (100%)	196 (100%)	800 (100%)	232 (100%)	793 (100%)
**Restored teeth**						
>10	78 (8.4%)	120 (11.7%) *^/F^	19 (9.8%) ^F^	99 (12.3%) *^/F^	35 (15.2%) *	85 (10.7%)
0–10	784 (85.2%)	839 (81.7%)	168 (86.6%)	650 (81%)	176 (76.2%)	663 (83.4%)
Cannot say	69 (7.4%)	67 (6.5%)	7 (3.6%)	54 (6.7%)	20 (8.7%)	47 (5.9%)
Total	931 (100%)	1026 (100%)	195 (100%)	803 (100%)	231 (100%)	795 (100%)

Data shown as number (percentage). *n* denotes the number of subjects. **^a^** Diet- and insulin- and/or metformin-treated GDM. **^b^** GDM diagnosed in previous pregnancy/pregnancies. **^c^** GDM diagnosed for the first time during current pregnancy. ^F^ Fisher’s exact test. Women with pharmacologically treated GDM compared to women with diet-treated GDM or women with recurrent GDM compared to women with first-onset GDM. ***** *p* < 0.05 compared to controls. **^†^** *p* < 0.001 compared to controls. **^‡^** *p* < 0.05 women with pharmacologically treated GDM compared to women with diet-treated GDM or women with recurrent GDM compared to women with first-onset GDM. **^§^** *p* < 0.001, women with pharmacologically treated GDM compared to women with diet-treated GDM or women with recurrent GDM compared to women with first-onset GDM. Categorical variables were determined by crosstabs and Pearson’s chi-square test and, where applicable, Fisher’s exact test, and parametric values were analyzed by independent samples *t*-test. “Cannot say” answers were removed from the analyses.

## Data Availability

The data supporting the findings of this study are available upon approval from the Finnish Institute for Health and Welfare, but restrictions apply to the availability of these data, which were used under license for the current study and so are not publicly available. The data contain registry data managed by the registration authority. The data are, however, available from the authors upon reasonable request and with the permission of the Ethics Committee of the Northern Ostrobothnia Hospital District and the Finnish Institute for Health and Welfare. Please contact Marja Vääräsmäki.

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
