# Peer review of "Increased Oral Care Needs and Third Molar Symptoms in Women with Gestational Diabetes Mellitus: A Finnish Gestational Diabetes Case–Control Study"

_ijerph, 2022, doi:10.3390/ijerph191710711_

Round 1
Reviewer 1 Report
This study assess whether GDM is associated with an increase in oral symptoms and a greater need for oral health care by comparing self-reported oral health and oral symptoms by women with GDM to those by non-GDM pregnant women.
The stated association oral care and third molar symptoms s in women with GDM are poorly supported by the data.
My main scientific concern is that the data are greatly over-interpreted.
If they stated the association, the significant improvement of GDM by the oral care, and tooth extraction would be desired.
Reviewer 2 Report
The authors conducted a multicenter case study to determine if women with gestational diabetes mellitus (GDM) are more likely to report increased need for oral healthcare or show increased oral symptoms during pregnancy. Pregnant women with or without GDM were surveyed by a questionnaire to determine their oral health and needs. The authors found that GDM is associated with an increased need for oral healthcare, removal of third molar and symptoms from third molar. The difference in third molar was mainly due to the parity and age of pregnancy, but the other associations were found mainly due to the presence of GDM. As GDM and oral problems constitute a vicious cycle, the authors suggest that more intensive oral healthcare and education should be implemented for pregnant and pre-pregnant women. The study may be limited given the ambiguous nature of some questions in the survey, but nevertheless provides useful support for the association between GDM and oral health given its large sample size.
Minor comments:
1. In the discussion, the authors cited a previous study that reported association between GDM and periodontitis and another study that had mixed results. The authors should also cite and comment on this study (https://pubmed.ncbi.nlm.nih.gov/31484379/) which reported that women with GDM showed better oral health parameters compared to women without GDM.
2. There seems to be an extra “5” in line 394.
3. The authors can consider changing the title to “Increased oral care needs and third molar symptom...” The current title may read like there is increased need for third molar symptoms in women with GDM.
Reviewer 3 Report
The manuscript entitled “Increased need for oral care and third molar symptoms in women with gestational diabetes mellitus: A Finnish Gestational Diabetes case-control study” performed a case-control study to evaluate the role the oral care in gestational diabetes (GDM). The authors recruited 1035 pregnant women with GDM and 935 pregnant women without GDM. The authors found that oral symptoms are more common among women with GDM, especially women with recurrent or pharmacologically treated GDM, and authors suggested oral care will be beneficial for these GDM pregnant women. This is a multicenter-based Finnish study and the outcome is outstanding. The manuscript has merit for publication. However, some queries should be addressed from the authors' side.
1. 1. There 1 is a huge statistical difference between the average ages of the cases and control groups, although authors considered it model 1, 2 and 3.
2. 2. There was also a difference in primiparous, preeclampsia between the cases and controls that may have an impact on the study’s outcome. Authors should adjust the outcome with primiparous, preeclampsia and other sociodemographic and clinicopathological data.
3. 3. It is unclear why oral care is needed more in Pharmacologically treated GDM, Diet-treated GDM, and Recurrent GDM than in controls.
4. 4. The introduction should be improved by discussing the oral problems in GDM, causes of more serious oral problems in Pharmacologically treated GDM, Diet-treated GDM, and Recurrent GDM.
5. 5. Minor grammatical errors were found that should be corrected.
Reviewer 4 Report
Thank you for the opportunity to review the paper: Increased need for oral care and third molar symptoms in women with gestational diabetes mellitus: A Finnish Gestational Diabetes case control study.
Pukkila et al. present a paper whose objective is: to evaluate whether gestational diabetes mellitus (GDM) is associated with a self-reported increased need for oral health care and increased oral symptoms, including third molar symptoms, during pregnancy.
I have some comments that could improve the paper.
In line 162 the authors describe that they will use the U-Mann-Whitney test to evaluate variables with a skewed distribution. However, there is evidence that with relatively large sample sizes, the use of the U-Mann-Whitney test may not be the most appropriate test. Any test in which the power of the test depends on the sample size is prone to overpowering. Donald W. Zimmerman (2003) A Warning About the Large-Sample Wilcoxon-Mann-Whitney Test, Understanding Statistics, 2:4, 267-280, DOI: 10.1207/S15328031US0204_03. My recommendation is to use the Student's t test.
In this same sense, lines 163 and 164 describe the use of Fisher's exact test in the case of small samples, however, to my knowledge, in none of the comparisons small group comparisons are made. For example, the comparison of "Induction of labor" in Table 1. Could the authors clarify this point?
In lines 184-187 the authors describe the following: "To study whether missing values ​​have an effect on results between different models, we performed additional multivariable regression analyzes excluding all those cases with missing variables from the analyses". However, it is not clear to me whether the authors made a new model without missing values, or simply ran the models without taking missing values ​​into account. In my opinion, in the regression models the missing values ​​are not and should not be taken into account.
Also in lines 187 and 188 it is described: "Correlations between different parameters were analyzed by Spearman's correlation test. A value of P < 0.05 was considered statistically significant". I did not see these analyzes reported in the manuscript.
A more detailed description of how the variables were selected in the models is required.
One of the most novel results of the study was finding that Third molar symptoms were more common among women with pharmacologically. However, I believe that the discussion of this interesting finding should be expanded. For example, there is the possibility that pharmacological treatment could be influencing this association.
Thanks again for the opportunity to review this work
